# Rapidly Progressing to ESRD in an Individual with Coexisting ADPKD and Masked Klinefelter and Gitelman Syndromes

**DOI:** 10.3390/genes13030394

**Published:** 2022-02-23

**Authors:** Ramón Peces, Carlos Peces, Rocío Mena, Emilio Cuesta, Fe Amalia García-Santiago, Marta Ossorio, Sara Afonso, Pablo Lapunzina, Julián Nevado

**Affiliations:** 1Servicio de Nefrología, Hospital Universitario La Paz, IdiPAZ, Universidad Autónoma, 28046 Madrid, Spain; rpecser@gmail.com (R.P.); marta.ossorio@salud.madrid.org (M.O.); sara.afonso@salud.madrid.org (S.A.); 2Area de Tecnología de la Información, SESCAM, 45071 Toledo, Spain; carlos.peces@sescam.jccm.es; 3Instituto de Genética Médica y Molecular (INGEMM), Hospital Universitario La Paz, IdiPAZ, Universidad Autónoma, 28046 Madrid, Spain; mariarocio.mena@salud.madrid.org (R.M.); feamalia.garcia@salud.madrid.org (F.A.G.-S.); plapunzina@gmail.com (P.L.); 4Servicio de Radiología, Hospital Universitario La Paz, IdiPAZ, Universidad Autónoma, 28046 Madrid, Spain; emilio.cuesta@salud.madrid.org; 5CIBERER, Centro de Investigación Biomédica en Red de Enfermedades Raras, ISCIII, 28046 Madrid, Spain; 6ITHACA, European Reference Network, Hospital Universitario La Paz, IdiPAZ, Universidad Autónoma, 28046 Madrid, Spain

**Keywords:** ADPKD, apoptosis, chronic kidney disease progression, fibrosis, Gitelman syndrome, intracranial aneurysm, Klinefelter syndrome

## Abstract

Autosomal dominant polycystic kidney disease (ADPKD) is the most common monogenetic hereditary renal disease, promoting end-stage renal disease (ESRD). Klinefelter syndrome (KS) is a consequence of an extra copy of the X chromosome in males. Main symptoms in KS include hypogonadism, tall stature, azoospermia, and a risk of cardiovascular diseases, among others. Gitelman syndrome (GS) is an autosomal recessive disorder caused by SLC12A3 variants, and is associated with hypokalemia, hypomagnesemia, hypocalciuria, normal or low blood pressure, and salt loss. The three disorders have distinct and well-delineated clinical, biochemical, and genetic findings. We here report a male patient with ADPKD who developed early chronic renal failure leading to ESRD, presenting with an intracranial aneurysm and infertility. NGS identified two de novo PKD1 variants, one known (likely pathogenic), and a previously unreported variant of uncertain significance, together with two SLC12A3 pathogenic variants. In addition, cytogenetic analysis showed a 47, XXY karyotype. We investigated the putative impact of this rare association by analyzing possible clinical, biochemical, and/or genetic interactions and by comparing the evolution of renal size and function in the proband with three age-matched ADPKD (by variants in PKD1) cohorts. We hypothesize that the coexistence of these three genetic disorders may act as modifiers with possible synergistic actions that could lead, in our patient, to a rapid ADPKD progression.

## 1. Introduction

Autosomal dominant polycystic kidney disease (ADPKD) is mainly caused by mutations in two main genes, *PKD1* (OMIM#: 601313) and *PKD2* (OMIM#: 173910) [1,2]. It is characterized by renal, progressive, age-related, fluid-filled cyst development, promoting ESRD [3,4]. It is widely known that phenotype variability is highly associated with the mutated gene (*PKD1* variants versus *PKD2* variants), and the type of the variant (frameshift/nonsense versus non-truncating variants) [5]. However, the existence of several modulator genes, and the existence of hypomorphic alleles may also largely affect the clinical features of ADPKD [5,6,7,8,9,10,11,12], including extra-renal clinical manifestations such as hepatic cysts, hernias, mitral valve prolapse, and intracranial aneurysms [3,4]. The renal disease progression of ADPKD involves not only cystogenesis, but also the activation of inflammatory and fibrotic pathways as a consequence of endothelial dysfunction [13,14]. Indeed, ADPKD, as a single gene disorder, is associated with numerous signaling cascades and reactions involved in cellular polarity, proliferation, fibrosis, and apoptosis [13,14,15]. Klinefelter syndrome (KS) is the most common genetic cause of infertility, and is the most variable aneuploidy (0.15% of the general population), showing high phenotypic variability and comorbidities. It is one of the common sex chromosome disorders in men. Interestingly, approximately 64% of these cases were undiagnosed throughout life [16,17]. A non-disjunction of paired X-chromosomes during the first or second meiotic division [17] may contribute to its origin, and is equally due to a paternal/maternal meiotic missegregation event. KS clinical manifestations were associated with increased risk for mitral valve prolapse, pulmonary embolism, aortic valvular disease, autoimmune diseases, the rupture of the berry aneurysms, lower-extremity varicose veins, deep vein thrombosis, venous stasis ulcers, type II diabetes mellitus and metabolic syndrome [18,19,20,21], extra-gonadal germ cell tumors and non-Hodgkin lymphoma [22], a 20-fold-higher risk of developing breast cancer, and osteoporosis. KS patients also have hypogonadism, which may increase the risk of atherogenesis [18]. However, the molecular mechanisms underlying the KS phenotype have not been clearly understood yet, although both genetic and epigenetic effects, due to the supernumerary X-chromosome, seem to contribute to this pathological pattern [23,24,25,26]. Gitelman syndrome (GS, OMIM #263800) is a rare autosomal recessive disorder due to homozygous or compound heterozygous variants in the SLC12A3 gene which encodes the thiazide-sensitive Na^+^/Cl^−^ cotransporter. GS is clinically characterized by hypomagnesemia, hypochloremic metabolic alkalosis, hypokalemia, hypereninemia, and hyperaldosteronism with normal or low blood pressure and salt loss [27]. Most of patients with GS present during childhood or early adulthood, however, it has high clinical variability, and some patients may not have GS features [27,28,29,30,31]. We here report a patient with ADPKD and coexisting masked KS and GS, based on genetic studies and clinical features. Although the three disorders may have different genetic, biochemical, and clinical findings, it is possible that the overlapping findings have influenced each other. We investigated the possibility that this fact may be a consequence of putative genetic interactions and/or additive or synergistic effects through the different signaling pathways.

## 2. Materials and Methods

### 2.1. Follow-Up Study

We compared the clinical evolution of ADPKD in the proband with our three control PKD1 defined cohorts by kidney magnetic resonance imaging (MRI) [32] and estimated glomerular filtration rate (eGFR) studies. To this end, total kidney volume (TKV), height adjusted total kidney volume (htTKV), and eGFR were determined. The eGFR was determined by the chronic kidney disease epidemiology collaboration (CKD-EPI) equation using the serum creatinine and the serum cystatin C values calculated using the formula: eGFRCysC = 77.24 × (Cys^−1.2623^). Mayo clinic imaging classification and the prediction of future eGFR based on the classification were determined in the proband [33].

### 2.2. Genetics Analysis

#### 2.2.1. ADPKD by MLPA Analysis

ADPKD gross deletion analysis was made by means of multiplex ligation-dependent probe amplification (MLPA) using the SALSA MLPA KIT P351/P352 for PKD1/PKD2 (MRC-Holland, Amsterdam, The Netherlands), following the manufacturer’s instructions. Data analysis was done with Coffalyser v6 MLPA Analysis Software (MRC-Holland).

#### 2.2.2. Massive Parallel Sequencing Analysis

Massive parallel sequencing analysis was performed with a custom-targeted next-generation sequencing (NGS) gene panel (Nefroseq v1.2) designed for the study of the full spectrum of genetic nephropathies (which includes 380 genes related to kidney diseases) [34]. The sequence was captured using SeqCap EZ technology (Roche Nimblegen. Madison, WI, USA) and subsequently ran on a Hiseq4000 (Illumina. San Diego, CA, USA). Bioinformatics analysis was performed in-house by using publicly available software tools (trimmomatic-0.32; Bowtie2 v2.1.0; Picard-tools v1.27; Samtools v0.1.19-44428cd; Bedtools v2.26.0; Genome Analysis TK v3.3-0 y SnpE 4.1l; ClinVar date 20140703 dbscSNV1.1 dbNSFP version 3.0 dbSNP v138), and simultaneously with variant Caller V2.1 tool (Illumina). In silico pathogenicity prediction was analyzed using Alamut 2.7 (interactive Biosoftware, Rouen, France) and others such as SIFT Ensembl 66; Polyphen-2 v2.2.2; Mutation Assessor, release 2; FATHMM, v2.3; Gerp2; PhyloP; CADD, v1.3. Population frequencies of the detected variants were assessed using the Exome Aggregation Consortium (ExAC; Exac r0.3) data; 1000 genome project; Spanish Exon Variant Project; NHLBI exome sequencing project: ESP6500_EA_AF). After the filtering of the relevant variants, validation of the candidate variants in the patient and parents was carried out through traditional Sanger Sequencing (see below). In parallel to the analysis process, exhaustive quality control of the sequenced samples was performed according to a custom pipeline (a suite of QC scripts that facilitate data quality assessment), available upon request. Classification and interpretation of the variants were made according to the American College of Medical Genetics and Genomics (ACMG) guidelines.

#### 2.2.3. Sanger Sequencing

*PKD1* mutation screening for coding sequences and intron/exon boundaries for exons 27 and 29 were confirmed by Long Range-PCR and direct sequencing. SLC12A3 mutation screening for coding sequences and intron/exon boundaries for exons 16 and 25 was performed by PCR and direct sequencing. PCR conditions and primers (designed with the help of Primer3 plus v04.0 Software) are available upon request. PCR products were sequenced using BrightDye Terminator cycle kit (Nimagen, Nijmegen, The Netherland) and run on an ABI3730XL Sequencer (Thermo Fisher, Waltham, MA, USA).

#### 2.2.4. Azoospermia Studies

Our pipeline routine for azoospermia studies includes microdeletions of the Y-chromosome, karyotyping, and cystic fibrosis analysis. The Y-chromosome microdeletions analysis was performed by standard fragment analysis, by multiplex PCR of STS markers and direct sequencing (primers are available upon request). The molecular diagnosis of Y-chromosome microdeletions was performed following the recommendations of the EAA/EMQN best practice guidelines [35]. Briefly, two multiplex PCR were carried out in each patient. Multiplex A contained primers for SRY and ZFX/ZFY genes and for AZF a, b, and c regions (sY86, sY127, and sY254, respectively). A second multiplex PCR was performed to reinforce diagnostic accuracy. This multiplex contained a second set of primers for AZF a, b, and c regions (sY84, sY134, and sY255, respectively), in addition to primers for SRY and ZFX/ZFY genes. In parallel to the patient’s DNA sample, a DNA sample from a normal male with normal spermatogenesis and a female were used as positive and negative controls, respectively. A water sample was also run as a control for reagent contamination. Cystic fibrosis screening was made by a commercial kit of CFTR analysis (Devyser; HQ, Hägersten, Sweden). Cytogenetic analyses were performed on peripheral blood lymphocytes by GTG-banded metaphases at a resolution of about 550 bands according to standard laboratory protocol using Chromosome Kit P (Euroclone, Siziano PV, Italy). Normally, 20 metaphases were counted.

## 3. Results

### 3.1. Case Presentation

The proband, a 36 year-old male German patient presented in our outpatient clinic for renal evaluation in April 2016. He was the fourth child of unrelated and healthy parents (Figure 1A).

He was born after an uncomplicated pregnancy. The family history was negative for abortion and genetic abnormalities. He was diagnosed with arrhythmia at 21, and epididymal cysts at 27 years old in Germany. At that age, abdominal ultrasonography (US) showed bilateral, enlarged, cystic, and echogenic kidneys being diagnosed with ADPKD. Surgical intervention of bilateral inguinal hernias was undertaken in Germany. At this time, the patient was also receiving intermittent treatment with Ramipril (5 mg/daily), which was finally discontinued. His serum creatinine levels were 2 mg/dL with an eGFR of 44 mL/min/1.73 m^2^. His father was alive at 77 years old without a history of ADPKD or any other renal disease. His mother died at the age of 51 (unrelated to genetic known cardiac/renal disorder). His three brothers were asymptomatic based on medical history. There was no history of intracranial aneurysms, ADPKD, or other renal diseases in other family members. Our first physical examination (in 2016, at 36 years old) established a well-developed male (weight, 85 kg; height, 196 cm; body mass index (BMI), 22.1 Kg/m^2^), presenting a decreased testicular volume. Blood pressure was 115/70 mmHg without any treatment. His serum creatinine level was 2.9 mg/dL and eGFR of 27 mL/min/1.73 m^2^. ECG showed a complete blockage of the right branch of the bundle of His, and abdominal MRI documented bilateral, enlarged, cystic kidneys. A TKV, 726 mL (right kidney 336 mL, left kidney 390 mL) and htTKV, 370 mL/m were measured by manual segmentation using MRI [32]. We classified the patient into Mayo class 1B [33]. However, the prediction of future eGFR based on Mayo imaging classification determined and estimated ESRD at 48 years old. No liver cysts were appreciated (Figure 1B). The patient also had a saccular cerebral aneurysm of the left anterior communicating artery measuring 11 × 5 mm, that we established by MRI angiography and confirmed by computed tomography (CT) angiography and arteriography (Figure 1C). An intent of embolization resulted in failure, and he denied surgical intervention of the aneurysm. He was married but without offspring. Due to couple infertility, the patient also underwent urologic evaluation in our Centre, which confirmed epididymal cysts and diagnosed azoospermia. Indeed, he had increased serum luteinizing hormone (LH) (24.70 IU/L; normal range: 1.50–9.30 IU/L) and follicle-stimulating hormone (FSH) levels (60.12 IU/L; normal range: 1.40–18.10 IU/L) with normal prolactin (15.66 ng/mL; normal range: 2.10–17.70 ng/mL) and total testosterone (4.49 ng/mL; normal range: 1.65–7.53 ng/mL), but low free testosterone levels (12.30 pg/mL; normal range: 15.0–50.0 pg/mL). His triiodothyronine (T3), thyroxin T4, and thyroid stimulating (TSH) hormones, and vitamin-D (48 ng/mL; normal range: 30–100 ng/mL) levels were normal, but parathyroid hormone (iPTH) (levels (169 pg/mL; normal range: 18.5–88.0 pg/mL) were increased. Bone mineral density examination also showed marked osteoporosis. A control CT cerebral angiography performed 18 months after the first observation (at 38 years old) revealed that the cerebral aneurysm of the left anterior communicating artery had increased up to 11.9 × 5.3 mm. At 42 years old, his serum creatinine level was 5.4 mg/dL and eGFR was 12 mL/min/1.73 m^2^ (Table 1). Several intents of treatment with a low dose of Telmisartan (10 mg/daily) resulted in hypotension and were discontinued.

### 3.2. Follow-Up Study

Since genetic studies showed a likely pathogenic missense mutation in *PKD1* in heterozygosis, we classified the patient as PROPKD 4 score (low risk of progression) [36] and Mayo class 1B (low risk of progression). However, his TKV and htTKV values were significantly smaller than two of our *PKD1* cohorts used to monitor our proband (*PKD1* patients ≤ 40 years, and *PKD1* with BMI < 25), but not smaller than *PKD1* without hypertension (Table 2). In fact, eGFR in the proband was significantly lower than the three *PKD1* cohorts, according with his prediction of future eGFR (ESRD at 48 years, high risk of progression). Moreover, Table 1 shows that the patient experimented a rapid progression from CRF to ESRD in only four years.

Interestingly, the proband had no clinical features related to GS, at all. He showed that the serum magnesium (1.6–2.0 mg/dL) and potassium levels (3.9–4.7 mmol/L), and venous (pH 7.31–7.41) were within normal limits, although HCO3- levels in blood were inappropriately higher (24–31 mmol/L) for his grade of CRF (Table 1). It is remarkable that during the follow-up course his blood pressure remained normal to low, despite the presence of severe CRF, including ESRD; he also had an intolerance to telmisartan treatment.

### 3.3. Genetic Analysis. Coexistence of PKD1 and SLC12A3 Variants, and 47, XXY Karyotype in Our Proband

#### 3.3.1. Fertility Studies

Cytogenetic analysis of peripheral blood showed a 47, XY,+X karyotype (Figure 1D). Y-chromosome microdeletion analysis and variant analysis of the *CFTR* gene were negative.

#### 3.3.2. NGS Studies

NGS studies detected a previously described missense variant in *PKD1*; NM_001009944.3:c.9499A>T(p.Ile3167Phe) (Figure 1E, left) in heterozygosis at exon 27. This variant was reported in different ethnicities [12,37,38,39], including European Non-Finnish (although in the latter ones, with a population frequency under the threshold established for nonpathogenic variants; source, Varsome). It was previously classified as a variant of uncertain significance VUS (http://pkdb.mayo.edu/ (accessed on 14 December 2021); ADPKD mutation database), and as a disease (Uniprot), which reflect conflicting interpretations of pathogenicity (ClinVar), as well as VUS (LOVD), VUS (HGMD), and, following ACMG/AMP ^2^ criteria [40], it is likely pathogenic (LP) (PM1, PP2, PP3, PP4, PP5, BS2). In addition, we detected a second *PKD1* missense variant NM_001009944.3:c.9756G>C(p.Glu3252Asp) (Figure 1E, right) in heterozygosis at exon 29, that was not previously reported in known databases (Mayo, Uniprot, ClinVar, LOVD, ALAMUT), nor in gnomAD research projects. It was classified using ACM/AMP ^2^ criteria as VUS (PM2, PP2, PP3). Both variants were validated by Sanger sequencing in the proband. In addition, NGS analysis also established two *SLC12A3* variants, NM_000339.3:c.1928C>T(p.Pro643Leu) (Figure 1F, left) in exon 16 and NM_000339.3:c.2891G>A(p.Arg964Gln) (Figure 1F, right) in exon 25, associated with GS. Both SNVs were previously described in clinical databases (p.Pro643Leu: ClinVar; conflicting interpretation of pathogenicity; LOVD, LP; Uniprot, disease; Varsome, LP/pathogenic (P) and p.Arg964Gln: ClinVar; LP/P; LOVD, P; Varsome, LP, P) and classified under ACMG/AMP 2 criteria as LP and P, respectively. Both variants were also confirmed by Sanger sequencing only in the proband (no other members of the family were available at this time). Thus, we cannot establish familial segregation for those and previous variants. No other pathogenic variants were found in other ADPKD-associated genes (*PKDH1, GANAB, ACE*, etc,) or related to cyst progression. The analysis of CNV by means of MLPA for *PKD1*, and *PKD2*, and the rest of the panel (using the algorithm Lacon v1.2 tool; INGEMM) was negative. Genetic findings are shown in Table 3.

## 4. Discussion

### 4.1. ADPKD

In ADPKD, the uncontrolled growth of renal cysts increases renal volume and the destruction of the parenchyma progressively, resulting in renal failure in most patients [41]. In addition, endothelial dysfunction which occurs very early in the course of the disease appears to be involved in increased oxidative stress and inflammation [42,43]. Then, the progression of ADPKD is highly variable, in part because of the gene affected [1,2,3,4,5,6,7], as well as the presence of other variants modifying its effect. In fact, how environmental factors and/or modifier genes/variants may modulate ADPKD, involving a rapid progression of the disease [6,7,8,13,44,45], is still unknown. Indeed, the presence of additional gene variants may promote cystogenesis and/or fibrosis, and potentiate the advancement toward ESRD [4,5,6,7,8,14]. At this point, there is some evidence of how synergistic effects between PKD1 and *PKD2* loci may accelerate the progression to ESRD approximately 20 years earlier than patients with only *PKD1* sequence variants [6,7].

Cyst growth in ADPKD is associated with increases in epithelial cell proliferation, dedifferentiation, and fluid secretion. Then, the enlargement of cysts affects surrounding nephrons interrupts kidney function significantly [36,41,44]. Indeed, cyst pressure induces renin–angiotensin–aldosterone system (RAAS) activation and kidney hypoxia. At the late stage of ADPKD, cyst formation is always accompanied by extracellular matrix deposition and fibrosis formation [14,15,41], which reduces renal function and eventually leads to progression to ESRD [14,41]. In addition, oxidative stress/reactive oxygen species (ROS) in ADPKD has also been considered a new player and/or early predictor for such disease progression [46,47], either directly, starting endothelial dysfunction and generating arterial atherosclerosis, or weakening of the arterial medial layer leading to arterial aneurysms [48]. Therefore, ADPKD can also be related to extra-renal manifestations, such as symptomatic extra-renal cysts, hypertension, and subarachnoid hemorrhage from intracranial aneurysms [49,50].

Interestingly, we classified our proband as low risk of progression (Mayo class 1B and PROPKD 4 score), however, the eGFR predicted established the development of ESRD at 48 years (rapid progression). In fact, his real situation of ESRD had been achieved at age of 40 (eGFR 16 mL/min/1.73 m^2^) or 42 years (eGFR 12 mL/min/1.73 m^2^). Furthermore, his eGFR was significantly lower than expected when it was compared with three different age-matched ADPKD cohorts, one with similar htTKV (Table 2). Thus, we suggest the predominant expression of renal tubulointerstitial inflammation, apoptosis, fibrosis, and/or vascular disease over cyst growth in this patient. It may explain the discordance observed between different markers of risk of ADPKD progression and between htTKV and eGFR evolution.

### 4.2. Klinefelter Syndrome

The presence of more than one X-chromosome in men characterized KS, which is the most common genetic cause of human male infertility. KS is related to high comorbidity with a lower life expectancy. It is thought that overexpression of some genes on the extra X-chromosome may be the cause. However, what genes and how they interact remain unclear [51,52,53,54]. We know that KS individuals may have an increased risk of cardiovascular and cerebrovascular diseases [18], an aspect that could affect the severity of the extra-renal complications in our patient. Due to the Lyonization phenomenon, in KS individuals only one of the X-chromosomes in the somatic cells is fully active, although around 15% of the genes on the inactive X-chromosome continue to be expressed, albeit at a lower level. Thus, recent works, focused on identifying those gene clusters that escape from X-inactivation in KS patients, have revealed altered functionality in KS by several protein complexes grouped by clusters [53]. Among those clusters, one of the most significant comorbidity clusters includes PKD1 [53]. In addition, KS can lead to structural alterations in tissues [16] in different ways, including heart diseases, insulin resistance [18], thrombotic events, or subarachnoid hemorrhage from an intracranial saccular aneurysm rupture [20,21,55]. These facts could affect the process of mitochondrial oxidative phosphorylation (OXPHOS) [24]. The association of KS with other disorders is expected due to the relatively high frequency of this entity [56].

### 4.3. Hypergonadotropic Hypogonadism

Male ADPKD patients may present with several reproductive system abnormalities and infertility [57,58,59]. Patients with KS usually show gonadal dysfunction with increased FSH and LH levels. Similar hormonal dysfunction was often found in male patients with CRF of any etiology [60]. So, CRF is associated with an increase in the FSH and the LH levels due to the impairment of the renal function in catabolizing these hormones, with normalization after renal transplantation [60]. Consequently, if a patient has these two conditions (KS and CRF from ADPKD) at the same time, the presence of an increase in the FSH and the LH levels, which are seen in CRF, can overlap with the KS manifestations masking and leaving the KS undiagnosed. This case illustrates an under-recognized lag in the diagnosis of hypergonadotropic hypogonadism, since the superposition of the two events is rare (decrease in free testosterone and increase in FSH and LH levels), indicating that in addition to KS, this effect may be caused by coexisting CRF.

### 4.4. Gitelman Syndrome

GS caused by SLC12A3 sequence variations is characterized by hypochloremic metabolic alkalosis, hypomagnesemia, RAAS activation with normal or low blood pressure, salt loss, and hypokalemia [27]. Identification of biallelic pathogenic variants in *SLC12A3* is necessary for establishing the diagnosis of GS. NGS showed two previously reported heterozygous missense variants in the *SLC12A3* gene: c.1928C>T(p.Pro643Leu) and c.2891G>A(p.Arg964Gln). However, we have not found electrolyte alterations characteristic of GS in our proband. Thus, the phenotypic effect of such missense variants is difficult to evaluate, and some pathogenic variants may not be sufficient to cause phenotypic changes related to GS [27,28,29,30,31], although, we cannot rule out that modifier genes are involved in the onset of GS or that phenotypic changes caused by this compound heterozygous would be apparent later in life. Indeed, our proband had a reduced eGFR at the time of the first evaluation, and this fact could in part mask the electrolyte alterations of GS (Table 1).

### 4.5. Co-Occurrence of ADPKD, KS, and GS

We describe the first report of ADPKD combined with KS and GS, which initially were unnoticed, indicating the need to consider the concurrent existence of additional disorders in cases of atypical ADPKD manifestations. This patient carries clinically significant variants in two different genes and a huge genomic rearrangement (one extra X-chromosome) responsible for additional pathology. Therefore, all these entities in one individual are not common, but it depends on the prevalence of each disease. Taking into account that their prevalences are around one in 500–1000 (ADPKD) [3,4] and one in 500–1000 (KS) male births [16,17], respectively, the probability for the simultaneous occurrence of these two disorders is approximately 1/250,000–1,000,000 births. Therefore, the occurrence of both ADPKD and KS in a single individual is in fact, theoretically, very rare. The prevalence of GS is estimated at approximately 1:40,000 [27]. Then, the occurrence of ADPKD, KS, and GS is even a rarer event. The effects of two or more genetic disorders in the same patient are unknown, and they can be additives, protectives, or neutrals [61], and may be influenced by the extent to which the clinical features with each individual disease overlaps with the other. Thus, the diseases can be distinct or overlapping. Distinct effects may affect different organ systems, whereas overlapping clinical effects are more likely to be caused by two genes encoding for proteins that normally interact within the same signaling pathway. Since ADPKD in association with KS and GS has never been reported in the literature, the possible effect of each other is currently unknown. Therefore, at this point, it is not clear if KS and GS will affect the cyst progression of ADPKD or vice versa. Indeed, the severe ADPKD phenotype denoted in our patient can be a consequence of two, not so strong variants in *PKD1* gene associated with other genetic variants (two *SLC12A3* variants and an extra X-chromosome). In addition, we cannot rule out that the *PKD1* variant c.9756G>C may have a role as a hypomorphic allele [11,12], or may occur from a putative alteration of topological chromatin organization (TAD; topologically associating domains). We now know that enhancers regulate the expression of distal target genes via long-range regulatory chromatin loops within the genomic context of TADs [62]. The later hypothesis needs to be also explored.

The clinical progression to ESRD and disease prognosis in this patient with ADPKD, KS, and GS (isolated KS or GS not producing CRF) could be potentially synergistic and comparable to individuals with variants in both *PKD1* and *PKD2* [6,7], hypomorphic *PKD1* variants and likely biallelic disease [12], the contiguous gene syndrome *TSC2-PKD1* [8], co-occurrence of ADPKD and hereditary renal hypouricemia [45], or dual sequence variants in *PKD2* and *COL4A1* [63]. In fact, our patient with an LP variant in *PKD1*, KS, and a compound heterozygous GS-associated variant showed a severe phenotype of ADPKD, characterized by the development of early CRF (at 27 years old) and rapid evolution to ESRD (at 40 years old) (Table 1), associated with an intracranial aneurysm (Figure 1C). He developed ESRD 15–20 years earlier than patients with a P/LP variant in *PKD1* (average age of 58 years) [4,24,37,38]. Indeed, our patient also carries features that are atypical for ADPKD, such as the development of early CRF without severe hypertension and quick evolution to ESRD, typically associated with significant cyst burden. However, he only manifested moderate ADPKD by kidney length. This fact suggests a putative effect of modifier genes with a predominant development of fibrosis over cyst growth, as has been demonstrated in a TGF-β1-induced fibrosis murine model [64,65]. Therefore, discordance between cyst burden and renal function decline trajectories that could be explained by lower cyst growth and remarkably slowed kidney enlargement in this ADPKD patient. In the present study, it was hypothesized that coexisting ADPKD with masked KS and GS may have a synergistic effect on the stimulation of rapid evolution to ESRD, and their synergistic effect may be the result of regulation of multiple signaling pathways, including vascular disease [66,67,68,69,70,71,72,73] (see Figure 2).

On the other hand, in both ADPKD and KS there is an elevated frequency of intracranial aneurysms and, in the case of associated variants in PKD1 protein and KS, it is expected to have at least additive intracranial aneurysms in disease presentation [21,49,50,55]. Thus, inferred by our data, he appears to be more severely affected than is typically reported with either condition alone. In fact, we detected an intracranial aneurysm with increased growth in a short period of time. Thus, the presence of an additive/synergistic effect may be explained by the underlying molecular function of the polycystin-1 protein and KS. Indeed, it has been previously established that loss of polycystin-1 function reduces control of Ras and aberrant activation of the mTOR pathway. Moreover, activation of the mTOR pathway and increased vascular endothelial apoptosis in KS [66] and ADPKD could affect the intracranial aneurysm growth in our patient [23,24,67]. Finally, other possible factors that could affect ADPKD progression and the growth of the intracranial aneurysm in this patient include also an increased RAAS stimulation, apoptosis, fibrosis, and vascular disease [26,27,68].

## 5. Conclusions

We present, for the first time a young ADPKD patient with an early, fast progression to ESRD, as well as growth of an intracranial aneurysm, accompanied with masqueraded KS and GS. We think that the presence of a 47, XXY karyotype and compound heterozygous variants in *SLC12A3* may increase the pathogenic action of *PKD1* variants. This is done, we presume, by increasing apoptosis, fibrosis, and producing predominant tubule–interstitial and vascular renal injury, instead of cystic growth, as it has been suggested in murine models. The use of NGS in the genetic diagnosis of different diseases is currently demonstrating the coexistence of several genetic variants in the same patient, which may contribute to the high variability and heterogeneity observed in numerous individuals. This case illustrates the importance of performing genetic testing to establish a complete diagnosis in ADPKD patients with atypical presentations and/or with fertility abnormalities.

## Figures and Tables

**Figure 1 genes-13-00394-f001:**
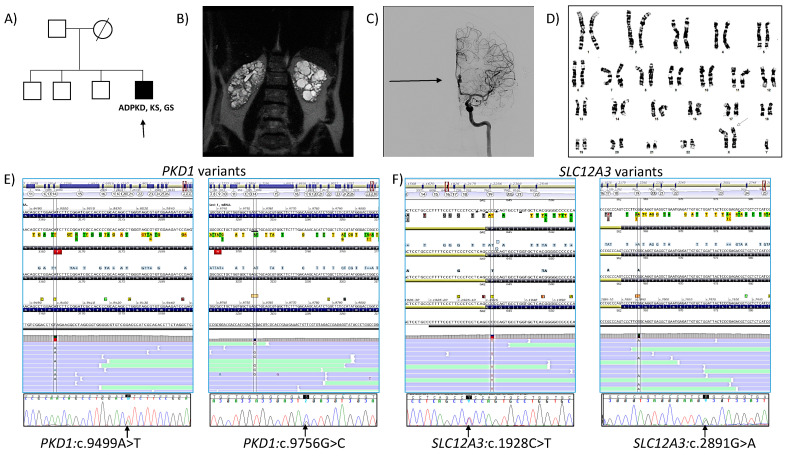
(**A**) Pedigree of the family. The proband (arrow). (**B**) Abdominal coronal T2 MRI of the proband at 36 years, showing a TKV of 726 mL (htTKV of 370 mL/m). (**C**) Cerebral angiography showing an aneurysm of the anterior communicating artery with hourglass image (arrow) (**D**) Cytogenetic analysis showing a 47, XXY karyotype, 50 metaphases are counted. (**E**) Molecular establishment of *PKD1* variants in the proband: NM_001009944.3:c.9499A>T and NM_001009944.3:c.9756G>C, by NGS (top) and Sanger sequencing (bottom). (**F**) Molecular establishment of *SLC12A3* variants in the proband: NM_0000339.3:c.1928C>T and NM_0000339.3:c.2891G>A, by NGS (**top**) and confirmed by Sanger sequencing (**bottom**). ADPKD, Autosomal dominant polycystic kidney disease; KS, Klinefelter syndrome; GS, Gitelman syndrome.

**Figure 2 genes-13-00394-f002:**
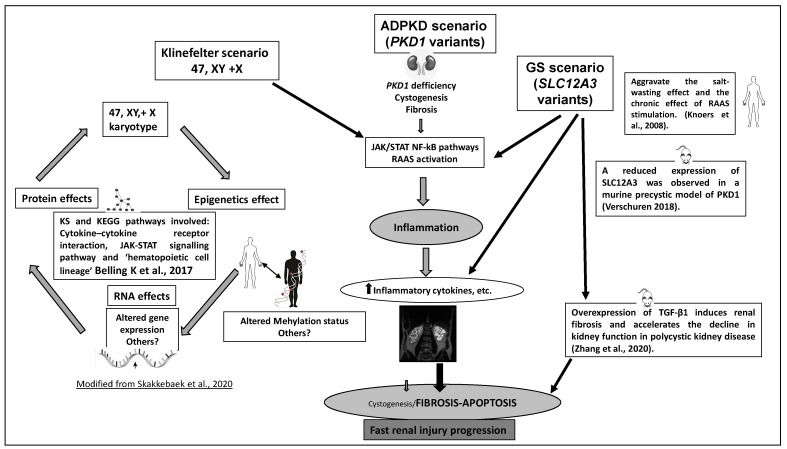
Hypothetical scheme for putative mechanisms on synergistic ADPKD pathogenesis and additional effects of KS and GS in our proband. Based in animal and human models of disease.

**Table 1 genes-13-00394-t001:** Evolution of electrolytes and renal function on the proband.

Age Years	Na mmol/L	K mmol/L	Cl mmol/L	Mg mg/dL	HCO_3_^−^ mmol/L	pH	Serum Creatinine mg/dL	eGFR mL/min/1.73 m^2^
36	142	4.7	109	ND	24	7.36	2.9	27
37	142	3.9	103	ND	30	7.31	3.4	21
38	143	3.9	105	2	29	7.41	3.9	18
39	138	4.3	104	2	31	7.35	4.2	16
40	144	4.2	110	1.8	24	7.31	4.3	16
42	143	4.1	106	1.6	27	7.33	5.4	12

Na, sodium; K; potassium, Cl, chloride; Mg, magnesium; HCO_3_^−^, bicarbonate ion; eGFR, estimated glomerular filtration rate; ND, not determined.

**Table 2 genes-13-00394-t002:** Clinical features of the proband when compared with three different *PKD1* cohorts in our Centre.

Parameter	Proband	*PKD1*’s Patients without HT (*n* = 23, Mean ± SD)	*PKD1*’s Patients ≤ 40 Year-Old (*n* = 58, Mean ± SD)	*PKD1*’s Patients with BMI < 25 (*n* = 54, Mean ± SD)
Age, years	36	35.0 ± 5.6	33.1 ± 6.3	38.1 ± 10.0
TKV, mL	726	662.0 ± 223.0	1301.0 ± 966.0	1264.0 ± 750.0
HtTKV, mL/m	370	382.0 ± 124.0	752.0 ± 563.0	746.0 ± 449.0
eGFR, mL/min/1.73 m^2^	27	137.0 ± 19.0	119.0 ± 30.0	117.0 ± 30.0

TKV, total kidney volume; htTKV, height adjusted total kidney volume; HT, hypertension; BMI, body mass index; eGFR; estimated glomerular filtration rate; SD, standard deviation.

**Table 3 genes-13-00394-t003:** Summary of the genetic findings that may be associated with clinical features on the proband.

Gene	Genomic Findings	Variant Type	Chromosome	Disease	Inheritance	Clinical Significance
*PKD1*	c.9499A>T (p.Ile3167Phe)	Missense	16p13.3	ADPKD	AD	Likely pathogenic
*PKD1*	c.9756G>C (p.Glu3252Asp)	Missense	16p13.3	ADPKD	AD	Uncertain significance
*SLC12A3*	c.1928C>T (p.Pro643Leu) c.2891G>A (p.Arg964Gln)	Missense Missense	16q13	Gitelman syndrome	AR	Pathogenic Pathogenic
X-chromosome	47, XY,+X karyotype	Aneuploidy	X	Klinefelter syndrome	−	Pathogenic

## Data Availability

The URLs for data presented herein are as follows: Exome Aggregation Consortium Browser, http://exac.broadinstitute.org/, accessed on 7 December 2021; The Genome Aggregation Database (gnomAD), https://gnomad.broadinstitute.org/, accessed on 7 December 2021; Human Genetic Variation Database, Exome Sequencing Project, http://evs.gs.washington.edu/EVS/, accessed on 7 December 2021; dbSNP Short Genetic Variations, http://www.ncbi.nlm.nih.gov/SNP/, accessed on 7 December 2021; The Genome Analysis Toolkit (GATK), https://www.broadinstitute.org/gatk/, accessed on 7 December 2021; Picard, http://broadinstitute.github.io/picard/, accessed on 7 December 2021; Annovar, http://annovar.openbioinformatics.org/en/latest/, accessed on 7 December 2021; Polyphen2, http://genetics.bwh.harvard.edu/pph2/, accessed on 7 December 2021; SIFT, http://sift.jcvi.org/, accessed on 7 December 2021; MutationTaster, http://www.mutationtaster.org/, accessed on 7 December 2021; RefSeq, http://www.ncbi.nlm.nih.gov/refseq/, accessed on 7 December 2021; UCSC Human Genome Browser, https://genome.ucsc.edu/, accessed on 7 December 2021; OMIM, http://www.omim.org/, accessed on 7 December 2021; ClinVar database, https://www.ncbi.nlm.nih.gov/clinvar, accessed on 7 December 2021; Beacon. https://beacon-network.org/#/1000/, accessed on 7 December 2021; Genomes Project. https://www.internationalgenome.org/category/phase-3kaviar, accessed on 7 December 2021; HGMD. http://www.hgmd.cf.ac.uk/ac/index.php, accessed on 7 December 2021; LOVD. https://www.lovd.nl/ Alamut^®^, accessed on 7 December 2021. https://www.interactive-biosoftware.com/alamut-visuai, accessed on 7 December 2021; Varsome. https://varsome.com/, accessed on 7 December 2021; ENCODE Project. https://www.encodeproject.org/, accessed on 7 December 2021.

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
