# Peer review of "Rapidly Progressing to ESRD in an Individual with Coexisting ADPKD and Masked Klinefelter and Gitelman Syndromes"

_genes, 2022, doi:10.3390/genes13030394_

Round 1
Reviewer 1 Report
In their Case report Peces et al., gathered data from an individual first followed in Germany then in Spain. While overall interesting, the case report is, in my opinion, out of order, not proof read and chaotic. One wonders the choice of subgroups (PKD1 vs the rest seemed more relevant), choice of details to support the conclusion that are too far fetch. The work presented as a case report, turns to a review and point of view which is damaging a potential interesting case.
A rigorous proof read of the manuscript should be done before submitting, on top of typos, legends are sometimes missing and a lack of homogeneity is to note (organization of the methods and results for example).
Here are my comments below (as following the document):
KS is THE most common genetic cause of infertility; the most variable aneuploidy (1/660, about .15% of the general pop).
Case presentation should be its own section, it is not really a material and methods (somehow a material), but It seems appropriate to leave a sole section dedicated to the case presentation and follow up performed here.
The legend of Figure 1 is incomplete: A/ pedigree of the family (no need for the rest), B/Abdominal coronal T2…F/ molecular establishment confirmed by NGS of SLC…the figure is also missing the Karyotype explanation, and precision on the number of Meiose analyzed. (i.e. are we in a mosaic case?).
Since the patient started is medical story in Germany and continue in spain, the authors need to be very concise. In this presentation the age are confusing (e.g., bilateral hernia in Germany, what age??)
Biometrics seems to be within standards at the first physical (age in spain?? I assume 36?). Precision on the Ramipril treatment?was there an evolution of the size of the cerebral aneurysm reported line 123?
Table 1 would gain in clarity if the standard value in Spain/ Germany could be added.
Table 2: these are not only clinical features, precision should be made on the ADPKD cohorts (are they PKD1, PKD2, mix???) for soundness of evaluation they should be PKD1 only, explanation on the below 40 group? Are they sorted by age at diagnosis or just age?
Comparison should be made to what is comparable, PKD1patients display highly variable phenotype, biased are induced if compared to PKD2…since the history of the patient is available, why not had the evolution of his parameters here as well?
Provide reference to the bioinformatics tools (Line 151)
Azoospermia studies requires a little more precisions: resolution (550 bands per haploid cells);
Results should be organized in the same order as the methods, this is too convoluted to follow.
Karyotype precisions are required : number of metaphases check (take into account Mosaicisms)??
Annotation Line 178 are not adequate: 0.0016 in exons; 0.005 in genes..
Sanger sequencing should be performed to see if the two PKD1 mutations are in cis or Trans, without the conclusion on this particular individual are uncertain.
Table 3/ a chromosome gain is not a mutation.
If the two mutations in PKD1 are on separate lanes and the two SLC12A3 are on the same, am I to understand that the mutations are in trans for PKD1 and in cis for SLC123????
At this stage of the manuscript (Line 208) I am still confused why the comparison of a PKD1 individual is performed with PKD2 patients. Hence, I don’t really know without further precision on the cohort chosen what is the value of the conclusion made by the Author. The patient genetics are of interest to appreciate the potential synergic effects of mutations on one another. Alas, the table presented and cohorts used for analysis are not sufficient (or selective enough) to support the discussion proposed and the cumbersome figure 2.
On that note, in the ADPKD paragraph, the author mentioned the fibrosis and inflammation of the patients, am I to understand biopsies were performed?
On the KS paragraph, I would not call 2 year a lower life expectancy, most KS individual are asymptomatic and are not diagnosed. The increased level of FSH should be mentioned in the Hypergonadotropic paragraph, line 284 the claim ‘illustrate an under-recognized pitfall in the diag of hypergonadotropic…” hypogono is in childhood, CRF in adulthood. Hence this is not a pitfall but a “lag” in diagnosis..superposition of the two events is rare
The rest of the discussion was hard to follow and prior to make further comments I would like to see a proof-read version, the 4.5 sound highly speculative in its current form. It might be a semantic issue; or not. At this stage I am lost between facts and hypothesis.
Minor comments ( but not that minor):
Typos and double space can be found throughout the manuscript (please, proof read before submitting; reviewing is a time consuming task performed for free by the scientific community, proof reading is a minimum to ask)
Refrain on reporting a man, but a patient instead. The mention of the mother’s suicide is not relevant and should be replace by death (unrelated to genetic known cardiac/ renal disorder).
Line 38 : modify “promoting” by “leading to”
Line 70: may not have GS features instead of may have not GS..
Line 74 remove ‘disorder in the’,…., that this fact may be a consequences (remove as) …
Line 79 : the proband, a 36-year old german man…instead of a german man is 36…
Line 89: he was diagnosed with ….at 21…
Line 90 and of epididymal cysts at 27 year.
These are few example, I cannot re write the full manuscript. Please do Proof read your manuscript.
Author Response
pen Review
Rev 1.
1.- In their Case report Peces et al., gathered data from an individual first followed in Germany then in Spain. While overall interesting, the case report is, in my opinion, out of order, not proof read and chaotic. One wonders the choice of subgroups (PKD1 vs the rest seemed more relevant), choice of details to support the conclusion that are too far fetch. The work presented as a case report, turns to a review and point of view which is damaging a potential interesting case.
.-Amended. A proofread has been made.
.-Case report description has been modified to try to avoid such a chaotic situation.
.-The control groups of ADPKD for comparison reasons have been modified and focus only on PKD1 as the reviewer suggested.
2.- A rigorous proof read of the manuscript should be done before submitting, on top of typos, legends are sometimes missing and a lack of homogeneity is to note (organization of the methods and results for example).
.-Amended. A proof read has been made.
.-Organization of Methods and Results have been modified, trying to improve them. The case report has been extracted from Methods and its description has been included in the Result section as an independent entity.
3.- KS is THE most common genetic cause of infertility; the most variable aneuploidy (1/660, about .15% of the general pop).
Those comments from the reviewer have been introduced within the text
4.- Case presentation should be its own section, it is not really a material and methods (somehow a material), but It seems appropriate to leave a sole section dedicated to the case presentation and follow-up performed here.
The case report and follow-up studies have been extracted from Methods and its description has been included in the Result section as an independent entity.
5.-The legend of Figure 1 is incomplete: A/ pedigree of the family (no need for the rest), B/Abdominal coronal T2…F/ molecular establishment confirmed by NGS of SLC…the figure is also missing the Karyotype explanation, and precision on the number of Meiosis analyzed. (i.e. are we in a mosaic case?).
Legend of Figure 1 has been modified according to reviewer´s suggestions. We included the description of karyotype and the number of the metaphases studied; 50. No mosaic is determined
6.-Since the patient started his a medical story in Germany and continues in Spain, the authors need to be very concise. In this presentation the age is confusing (e.g., bilateral hernia in Germany, what age??)
.-Amended, we try to chronologically separate from both countries the findings and intervention on the patient.
7.- Biometrics seems to be within standards at the first physical (age in Spain?? I assume 36?). Precision on the Ramipril treatment? was there an evolution of the size of the cerebral aneurysm reported in line 123?
In Spain, we started to follow him at 36 years old, in 2016. Cerebral aneurysm evolves in size between 36 and 38 years old, increasing from 11 x 5 mm up to 11.9 x 5.3 mm. Unfortunately, we do not have data or Images of reference from Germany, about the size of the aneurism when the Ramipril treatment was applied.
8.-Table 1 would gain in clarity if the standard value in Spain/ Germany could be added.
Table 1 includes only data from visiting our clinic, starting at 36 years-old
9.-Table 2: these are not only clinical features, precision should be made on the ADPKD cohorts (are they PKD1, PKD2, mix???) for soundness of evaluation they should be PKD1 only, explanation on the below 40 group? Are they sorted by age at diagnosis or just age?
.-Amended, we removed data from the other cohort regarding PKD2, we included in this table data from patients with PKD1 P/LP variants. The group from PKD1 with below 40 group? This means that the age at diagnosis and evolution are under the age of 40 years-old
A comparison should be made to what is comparable, PKD1patients display highly variable phenotype, biased are induced if compared to PKD2…since the history of the patient is available, why not had the evolution of his parameters here as well?
Data evolution of the proband is also included in table 2 side by side with those cohorts of the patients with PKD1 variants.
10.-Provide reference to the bioinformatics tools (Line 151)
.-Bioinformatics tools are now included in the text, Methods and their web URLS included after the bibliography.
11.-Azoospermia studies require a little more precisions: resolution (550 bands per haploid cells)
.-Amended, Azoospermia studies have included with a major resolution
12.- Results should be organized in the same order as the methods, this is too convoluted to follow.
Amended, Results are organized as the Method section was
13.- Karyotype precisions are required: the number of metaphases check (take into account Mosaicisms)??.
.-Amended, a major precision of the Karyotyping description has been include in Methods.
14.- Annotation Line 178 are not adequate: 0.0016 in exons; 0.005 in genes.
.-This sentence has been removed to avoid misunderstandings
15.-Sanger sequencing should be performed to see if the two PKD1 mutations are in cis or Trans, without the conclusion on this particular individual are uncertain.
Sanger sequencing has been performed already only in the proband. No other family members´ DNA is available for study. Thus, we cannot establish familial segregation for those and SLC variants.
16.- Table 3/ a chromosome gain is not a mutation.
Amended, this has been corrected.
17.-If the two mutations in PKD1 are on separate lanes and the two SLC12A3 are on the same, am I to understand that the mutations are in trans for PKD1 and in cis for SLC123????
Sanger sequencing has been performed already only in the proband. No other family members´ DNA is available for study. Thus, we cannot establish familial segregation for those and SLC variants.
18.- At this stage of the manuscript (Line 208), I am still confused why the comparison of a PKD1 individual is performed with PKD2 patients. Hence, I don’t really know without further precision on the cohort chosen what is the value of the conclusion made by the Author. The patient genetics are of interest to appreciate the potential synergic effects of mutations on one another. Alas, the table presented and cohorts used for analysis are not sufficient (or selective enough) to support the discussion proposed and the cumbersome figure 2.
-Amended, we removed data from the other cohort regarding PKD2, we included in this table data from patients with PKD1 P/LP variants. The group from PKD1 with below 40 group? This means that the age at diagnosis and evolution are under the age of 40 years-old.
On that note, in the ADPKD paragraph, the author mentioned the fibrosis and inflammation of the patients, am I to understand biopsies were performed?
No biopsies were performed at this point. We change the legend of figure 2 to better comprehension, we speculated this based on the murine experimental data
19.-On the KS paragraph, I would not call 2 year a lower life expectancy, most KS individual are asymptomatic and are not diagnosed. The increased level of FSH should be mentioned in the Hypergonadotropic paragraph, line 284 the claim ‘illustrate an under-recognized pitfall in the diag of hypergonadotropic…” hypogono is in childhood, CRF in adulthood. Hence this is not a pitfall but a “lag” in diagnosis..superposition of the two events is rare
Amended, comments have been incorporated within the text
20.-The rest of the discussion was hard to follow and prior to make further comments I would like to see a proof-read version, the 4.5 sound highly speculative in its current form. It might be a semantic issue; or not. At this stage I am lost between facts and hypothesis.
We try to edit such paragraph, including some comments.
Minor comments (but not that minor):
21.- Typos and double space can be found throughout the manuscript (please, proofread before submitting; reviewing is a time-consuming task performed for free by the scientific community, proofreading is a minimum to ask).
You are right. A proof-read has been performed.
22.-Refrain on reporting a man, but a patient instead. The mention of the mother’s suicide is not relevant and should be replaced by death (unrelated to genetic known cardiac/ renal disorder).
.-Amended, we are agreed. We change the sentences regarding suicide, the term patient is more appropriate.
23.-Line 38: modify “promoting” by “leading to”
.-Amended.
24.-Line 70: may not have GS features instead of may have not GS..
.-Amended.
25.-Line 74 remove ‘disorder in the’,…., that this fact may be a consequence (remove as) …
.-Amended.
26.-Line 79 : the proband, a 36-year old german man…instead of a german man is 36…
.-Amended.
27.-Line 89: he was diagnosed with ….at 21…
.-Amended.
28.-Line 90 and of epididymal cysts at 27 years.
.-Amended.
These are a few examples, I cannot re-write the full manuscript. Please do Proof read your manuscript.
Reviewer 2 Report
In the paper entitled “Rapidly progressing to ESRD in an individual with coexisting ADPKD and masked Klinefelter and Gitelman syndromes”, the authors describe the case of a man with ADPKD, who developed early chronic renal failure, presenting an intracranial aneurysm and infertility. DNA sequencing analysis identified two de novo PKD1 variants, one known (likely pathogenic) and an unreported VUS, together with two SLC12A3 pathogenic variants. In addition, cytogenetic analysis showed a 47, XXY karyotype.
The authors speculate that the presence of a 47, XXY karyotype and heterozygous variants in SLC12A3 may increase the pathogenic action of PKD1 variants, likely by increasing apoptosis, fibrosis and producing tubulointerstitial and vascular renal injury. However, The effect of the concomitant presence in the same patient of two or more genetic disorders is yet unknown, and they can be additives, synergistic, protectives or neutrals. This case exemplifies the importance to perform genetic testing to establish a complete diagnosis in ADPKD patients with atypical presentations.
The paper is well-written and thought-out. The clinical case is interesting and monitoring of the clinical status of the patient will help to dismantle many yet unanswered questions.
Author Response
Reviewer2
1.-In the paper entitled “Rapidly progressing to ESRD in an individual with coexisting ADPKD and masked Klinefelter and Gitelman syndromes”, the authors describe the case of a man with ADPKD, who developed early chronic renal failure, presenting an intracranial aneurysm and infertility. DNA sequencing analysis identified two de novo PKD1 variants, one known (likely pathogenic) and an unreported VUS, together with two SLC12A3 pathogenic variants. In addition, cytogenetic analysis showed a 47, XXY karyotype.
The authors speculate that the presence of a 47, XXY karyotype and heterozygous variants in SLC12A3 may increase the pathogenic action of PKD1 variants, likely by increasing apoptosis, fibrosis and producing tubulointerstitial and vascular renal injury. However, The effect of the concomitant presence in the same patient of two or more genetic disorders is yet unknown, and they can be additives, synergistic, protectives or neutrals. This case exemplifies the importance to perform genetic testing to establish a complete diagnosis in ADPKD patients with atypical presentations.
We agreed, genomic tools are essential today, to perform genetic testing to establish a complete diagnosis in ADPKD patients and other ones with atypical presentations.
2.-The paper is well-written and thought-out. The clinical case is interesting and monitoring of the clinical status of the patient will help to dismantle many yet unanswered questions.
Thanks. We agreed, follow-up data will help us to unravel this case.
Round 2
Reviewer 1 Report
The updated manuscript shows a great improvement in the presentation and discussion of the proband.
One comment though; Since no study were done to decipher if the mutations are in cis or trans, The authors should acknowledge the potential effect of the PKD1 mutations if in cis (eg; modification of TAD). This could also explain the peculiar ADPKD phenotype, unrelated to the two other diseases. (an hypothesis not really explored by the study).
ps: still not a fan of fig 2. Too messy, using conclusions both from models (murine)and patients.
Author Response
Answers to Reviewer 1, Round 2
Thanks to the reviewer for her/his valuable comments. we have tried to improve the style moving some paragraphs, including abbreviatures, etc.
Regarding the comments:
The updated manuscript shows a great improvement in the presentation and discussion of the proband.
-Thanks to the referee for the comment.
One comment though; Since no study were done to decipher if the mutations are in cis or trans, The authors should acknowledge the potential effect of the PKD1 mutations if in cis (eg; modification of TAD). This could also explain the peculiar ADPKD phenotype, unrelated to the two other diseases. (an hypothesis not really explored by the study).
.-A paragraph regarding this putative hypothesis is now included.
ps: still not a fan of fig 2. Too messy, using conclusions both from models (murine)and patients.
- we try to improve figure 2 to separate animal y human models to describe our hypothesis.